# Complementary Sparsity: Accelerating Sparse CNNs with High Accuracy on General-Purpose Computing Platforms

**Kang Zhao**[*]                                                                 *zhaokang29@huawei.com*
*Huawei Noah Ark's Lab*

**Yijun Tan**[*]                                                                 *tanyj1998@gmail.com*
*SKL of Processors, Institute of Computing Technology, CAS*

**Kai Han**                                                                       *kai.han@huawei.com*
*Huawei Noah Ark's Lab*

**Ting Hu**                                                                       *huting35@huawei.com*
*Huawei Noah Ark's Lab*

**Hanting Chen**                                                                  *chenhanting@huawei.com*
*Huawei Noah Ark's Lab*

**Tao Yuan**                                                                      *yuantao38@huawei.com*
*Huawei Noah Ark's Lab*

**Yunhe Wang**[†]                                                                 *yunhe.wang@huawei.com*
*Huawei Noah Ark's Lab*

**Jun Yao**[†]                                                                    *yaojun97@huawei.com*
*Huawei Noah Ark's Lab*

**Reviewed on OpenReview:** *https://openreview.net/forum?id=g1B4qgOw79*

## Abstract

Model sparsity is a promising approach to reducing parameters and FLOPs of convolutional neural networks (CNNs). Compared to unstructured or coarse-grained structured sparsity, fine-grained structured sparsity, e.g., N:M sparse pattern, can achieve better balance between accuracy and efficiency on general computing platforms like CPUs and GPUs. In particular, the 2:4 sparsity can accelerate CNN inference by $2\times$ speed and with negligible accuracy drop. However, N:M sparsity needs to be supported by GPU within specific hardware circuits and hardly achieve significant speedups on common GPUs. To accelerate CNNs with general-purposed computing resources and simultaneously retain the model accuracy as much as possible, this paper proposes complementary sparsity (CS). CS denotes that only one weight can be retained for weights spaced at the same distance. On the one hand, CS features high mask flexibility, which is naturally favorable to high model accuracy. Moreover, we propose a CS-specific sparse training method to improve CS-based CNNs' accuracy under high parameter sparsities ($>75\%$). On the other hand, CS itself is memory-access balanced and robust to pattern hyperparameters, making it an ideal candidate for speeding up CS-based convolution computation on CPUs and common GPUs. We thus propose a CS convolution parallel computing algorithm that adapts to common GPUs without sparse tensor cores. Experimental results show that compared to other sparsity patterns, the proposed CS achieves the optimal trade-off in terms of accu-

---

[*]Equal contribution.
[†]Corresponding author.

racy and latency for CPUs and common GPUs, respectively. Codes will be available at https://gitee.com/mindspore/models/tree/master/research/cv/CS.

# 1 Introduction

Weight sparsification is a crucial method to compress CNNs. The rationality behind weight sparsification is that there are redundant weights in regular CNNs which tend to generate overlapped features (Hoefler et al., 2021; Ayinde et al., 2019). Thus, removing a certain amount of weights in CNNs has little or manageable impact on CNN's accuracy, while it can significantly lower CNN's number of floating-point operations (FLOPs) during inference.

According to the extent of pruning freedom and acceleration affinity, the existent weight sparsification technologies for CNNs can be divided into three categories: unstructured sparsity (US), coarse-grained structured sparsity (CSS), and fine-grained structured sparsity (FSS). US, depicted in Fig. 1(a), also called random sparsity in some studies (Huang et al., 2022), permits pruning weights anywhere inside a weight tensor (Zhu & Gupta, 2017; Gale et al., 2019; Mostafa & Wang, 2019; Evci et al., 2020; Kusupati et al., 2020; Liu et al., 2021; Ma et al., 2021; Peste et al., 2021; Tai et al., 2022; Jaiswal et al., 2022; Park & No, 2022; Chen et al., 2021; Li et al., 2022a). Due to US's highest degree of pruning freedom, the most important weights affecting network quality can always be retained under any sparsity. Hence, unstructurally sparsed networks can preserve a decent accuracy even if sparsity is very high ($\geq 90\%$). However, the nonuniformity of weight distribution makes it nearly impossible to accelerate US-based convolution on general-purpose computing platforms. In contrast, for CSS, e.g., Fig. 1(b)-(d), the pruning granularity is block, channel, or filter -wise (Wen et al., 2016; Li et al., 2016; Gray et al., 2017; Ji et al., 2018; Liu et al., 2017; Tang et al., 2020; Ding et al., 2021; Hou et al., 2022; Liu et al., 2022; Chen et al., 2022; Zhang et al., 2022; He & Xiao, 2023). CSS's patterns are generally with high regularity, which can significantly speedup the network inference. For some of CSS patterns such as filter pruning and channel pruning, i.e., Fig. 1(c) and 1(d), the pruned CNNs can directly operate on the original platforms without any new acceleration algorithm design. Nonetheless, the relatively bigger pruning granularity inevitably entails that many important weights are removed together with unimportant weights. Under the same accuracy, CNNs within CSS own a lower compression ratio compared to that within US patterns.

In this study, we focus on FSS since it generally results in better tradeoffs between accuracy and efficiency. We classify a sparsity pattern as FSS if its pruning granularity is vector-wise (Yao et al., 2019; Mishra et al., 2021; Huang et al., 2022; Tan et al., 2022; Meng et al., 2020), e.g., Fig. 1(e)-(h). Compared with US and CSS, FSS possesses both high prunability and decent acceleration affinity. However, the existing FSS patterns more or less have some shortcomings, as shown in Table 1. N:M sparsity, which has been the most popular FSS pattern lately, mainly facilitates inference efficiency on specific hardware, e.g., Amphere architecture within sparse tensor cores (Choquette & Gandhi, 2020). Some work tries to accelerate N:M-base convolution on common GPUs without sparse tensor cores (Yao et al., 2019), but the practical speedup benefits compared to the dense convolution are probably limited, since the dense convolution has been adequately optimized and perfectly supported by current common GPUs. Shfl_BW Huang et al. (2022) and OVW Tan et al. (2022) sparse patterns achieve practical speedups on common GPUs, but CNNs using these two patterns have not reached a satisficatory accuracy so far.

In this paper, we propose a complementarily sparsed pattern to better balance the accuracy and inference efficiency of sparse CNNs. **The design principle behind complementary sparsity (CS) is to leverage the non-conflict strengths of the prior sparse patterns as much as possible while addressing their limitations**. Firstly, like N:M sparsity, CS prunes weights inside a vector. Secondly, the positions of the pruned weights and retained weights are complementary. For example, Fig. 1(e) shows a 50% CS. In this subfigure, a vector's shape is $8 \times 1$, as marked by the red frame. The 'complementary' means that inside the vector, if the 1st weight is pruned, then the 5th weight has to be retained. This logic is the same for the 2nd and 6th weights, and so forth. Lastly, the size of a minimum vector to form CS is variable, which property is called hyperparameter robustness. The hyperparameter robustness of CS makes the pattern very adaptive to different computing platforms, such as CPUs and GPUs.

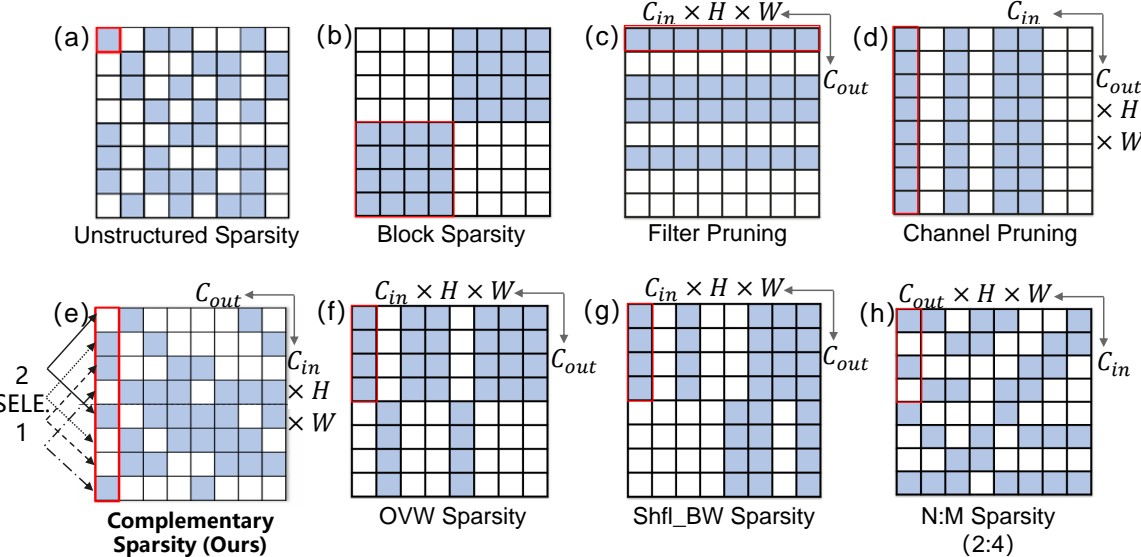

Figure 1: Visualization of different sparse patterns at the 50% sparsity. (a) Unstructured structured sparsity that allows to discard weights of arbitratry positions. (b)-(d) Coarse-grained structured sparsity. (e)-(h) Fine-grained structured sparsity. In particular, (e) is the proposed complementary sparsity and '2 SELE. 1' represents retaining one from two weights in all the complementary positions.

The major contributions of this paper are as follows:

- We propose a new sparse pattern—CS, which features both high mask flexibility and high acceleration affinity. CS allows pruning CNNs with less accuracy reduction and accelerating sparse CNNs on both CPUs and common GPUs.

- Used in the training phase, a CS-specific sparse training method is proposed to boost CS-based CNNs' accuracy under high sparsities. With the proposed method, CS-based CNNs perform on par with or better than that with N:M sparsity in terms of accuracy. At the 50% sparsity on ImageNet, CS-based ResNet50 achieves 76.37% accuracy with negligible accuracy loss. At the 93.75% sparsity, CS-based ResNet50 achieves 71.07% accuracy with a 5.32% drop, which is better than N:M sparsified ResNet50 which drops 5.8%.

- Used in the inference phase, a parallel acceleration algorithm is proposed to speedup the CS-based convolutions on common GPUs. With the acceleration algorithm, CNNs within CS achieves 2.59×~3.07× speedups at the 93.75% sparsity over the dense counterparts supported by cuDNN.

Through the algorithm-software co-optimization, the proposed CS reaches better tradeoffs between sparse CNNs' model quality and inference efficiency compared with other fine-grained structured sparse patterns. To be clear, the advantages of our CS over similar works are shown in Table 1.

## 2 Related Work

**Unstructured sparsity (US)** Neural networks within US have been researched for a long time. The winning ticket hypothesis denotes that there always exists a sparse neural network inside a dense network and the subnetwork can be trained to reach the comparable accuracy as the dense one (Frankle & Carbin, 2018). Since the hypothesis was proposed, a surge of studies have focused on developing good pruning methods to form US. Gale et al. (2019); Zhu & Gupta (2017) improve the magnitude-based pruning methods simply by gradually sparsifying and prolonging the training time, respectively. Rather than fully training a dense network before pruning, Mostafa & Wang (2019); Evci et al. (2020); Ma et al. (2021) adopt the

| Pattern | Acceleration w/o ASICs | Accu. |
|---|---|---|
| US | Almost impossible | High |
| Block sparsity | Yes | Low |
| Filter pruning | Yes | Low |
| Channel pruning | Yes | Low |
| N:MMishra et al. (2021) | Hard | High |
| Shfl_BWHuang et al. (2022) | Yes | Medium |
| OVWTan et al. (2022) | Yes | Medium |
| **CS (Ours)** | **Yes** | **High** |

Table 1: Comparison among different sparse patterns.

sparse training to directly generate the unstructured sparse neural networks. These sparse training methods basically contain a common mechanism that periodically prunes and regrows some weights according to some criterion. Furthermore, Peste et al. (2021) alternatively conducts sparse and dense training. In this way, both dense and unstructured sparse neural networks are generated after training. Instead of pruning weights by carefully designed criterion, Kusupati et al. (2020); Tai et al. (2022) learn the sparse masks by differentiation. In particular, the proposed method in Tai et al. (2022) reports the state-of-the-art accuracy of US-based CNNs on the ImageNet dataset. Generally, CNNs within US can not obtain significant speedup gains on CPUs and common GPUs due to the severe memory-access conflict.

**Coarse-grained structured Sparsity (CSS)** Normally, CSS can be enforced along either the input or output axes of a weight tensor. Separately shrinking the output and input axis is called filter and channel pruning, respectively. Pruning a square block in both input and output axes is called block sparsity (Gray et al., 2017). To our knowledge, Wen et al. (2016) is the first to propose filter pruning or channel pruning for CNN compression. In their study, the group LASSO method is directly enforced into weights to induce sparsities among filters or channels. Some studies like Liu et al. (2017); Ding et al. (2021) employ extra indicators to evaluate the filters, e.g., scaling factors in batch normalization layers, or properly placed $1 \times 1$ convolutions that can be absorbed during inference. Since pruning filters also result in pruning the related channels in the next layers, the proposed method in Li et al. (2016) jointly considers the impact of filter and channel pruning on network accuracy. Rather than developing various hypotheses to measure the importance of filters, Tang et al. (2020) assesses filters by observing the network responses to real data and adversarial samples. Besides, some principles originally used for US have lately been introduced to realize CSS, e.g., Hou et al. (2022); Tai et al. (2022). Despite these efforts, CSS-based CNNs' accuracy is still relatively lower and drops drastically especially when the required sparsity$> 70\%$.

**Fine-grained structured sparsity (FSS)** N:M sparsity is a well-known fine-grained structured sparse pattern where at most $N$ non-zero weights are retained for every continuous $M$ weights (Yao et al., 2019; Mishra et al., 2021; Lu et al., 2023; Zhang et al., 2023). However, N:M sparse pattern needs to be supported by GPUs embedded with sparse tensor cores. On common GPUs, the pattern hardly outperforms dense convolutions supported by cuDNN. To tackle the problem, Shfl_BW and OVW sparsity regard a $M \times 1$ vector as an entirety which is pruned or retained together (Huang et al., 2022; Tan et al., 2022). By this design, the retained weights and the related features during convolution computation can be easily indexed. Thus, Shfl_BW and OVW sparsity can accelerate convolutions on common GPUs to a great extent. However, the relatively large pruning unit of $M \times 1$ still decreases the flexibility (Hubara et al., 2021), which results in reduced model accuracy. In contrast, our CS can help maintain similar or better model accuracy relative to N:M sparsity, as well as achieve the practical speedups of sparse convolutions on common GPUs and CPUs.

**GPU acceleration for convolutions** So far, there are basically four sorts of parallelable algorithms to implement convolutions: direct convolution, Winograd (Chikin & Kryzhanovskiy, 2022), FFT (Wang et al., 2020), explicit general matrix multiplication (GEMM) (Jiang et al., 2022) and implicit GEMM (Zhou et al., 2021b). Among these, GEMM-base convolution algorithms are more performant on GPUs since the modern parallel computing platforms have highly optimized the GEMM operations (Chetlur et al., 2014; Jorda et al., 2019; Li et al., 2022b). However, explicit GEMM-based convolutions need to firstly invoke img2col to change

tensors to matrices, which is memory access-intensive and time-consuming. By contrast, implicit GEMM-based convolutions remove the memory access overheads, which is top-performed in most cases. Moreover, Tan et al. (2022) employs the implicit GEMM to accelerate sparse convolutions, which implies the potential of implicit GEMM to speedup other sparse patterns. In this work, the implicit GEMM is also utilized to develop the parallel acceleration algorithm of CS-based convolutions on common GPUs.

## 3 Method

### 3.1 Complementary Sparsity Formulation

For a sparse weight tensor $W$ with the shape of $C_{out} \times C_{in} \times F_h \times F_w$, each filter $W_i$ has the shape $C_{in} \times F_h \times F_w$ and is flattened. We use $W_i[j]$ to denote the $jth$ weight in the flattened $W_i$. $S$ is the sparsity of the weight tensor. Two key parameters are introduced to conveniently describe CS: 1) $K$. $K$ denotes the amount of the complementary positions from which only one single weight should be selected. For instance, at the 50% sparsity, there should be $K = 2$ complementary positions for selecting a weight. for the 75% sparsity, there are $K = 4$ complementary positions from which a weight is selected. 2) $M$. $M$ represents the address offset with which weights in the complementary positions can mutually be located. Specifically,

$$K = \frac{1}{1 - S} \tag{1}$$

$$M = \frac{L}{K}, L \in \{C_{in}/c, C_{in} * F_h * F_w\} \tag{2}$$

In Equation 2, if $L = C_{in}/c$, the typical values of $M$ include $2, 4, 8, 16$. Here $c$ only means that $C_{in}$ should be divisible by $M$. Then, a sparse weight tensor is regarded as complementarily sparsed as long as for any $W_i[j], i \in [1, C_{out}], j \in [1, M]$,

$$\|W_i[j], W_i[j + 1 * M], ... W_i[j + (K - 1) * M]\|_0 < K \tag{3}$$

Furthermore, a sparse weight tensor is regarded as strictly conforming to the pattern of CS if and only if

$$\|W_i[j], W_i[j + 1 * M], ... W_i[j + (K - 1) * M]\|_0 = 1 \tag{4}$$

To intuitively understand the definition of CS, Fig. 2 gives some examples strictly obeying the pattern of CS across multiple sparsities. Accordingly, the parameter values, i.e., $K$ and $M$ of each example, are listed in Table 2. Note that $K$ is only related to the specified sparsity $S$ and is independent of the specific weight tensor shapes. Unless otherwise specified, the rest of the paper only discusses the strict CS.

Figure 2: Examples of CS at different sparsities. The blue shading of (b)-(e) indicates the selected weights. (a) a dense weight tensor. For convience, the tensor is 2D and with the shape of $1 \times 16$, i.e., $C_{out} = 1$ and $C_{in} * F_h * F_w = 16$ (b) The 50% CS of the weight tensor. (c) The 75% CS of the weight tensor. (d) The 87.5% CS of the weight tensor. (e) The 93.75% CS of the weight tensor.

Table 2: Parameter values of the examples shown in Fig. 2.

| Sparsity (%) | $K$ | $M$ | Encoding bit number | Encoding results |
|---|---|---|---|---|
| 50 | 2 | 8 | 1 | 0,1,1,0,0,0,1,1 |
| 75 | 4 | 4 | 2 | 1,2,3,0 |
| 87.5 | 8 | 2 | 3 | 7,4 |
| 93.75 | 16 | 1 | 4 | 14 |

### 3.2 CS-specific Gradual Sparse Training Method

The aim of designing a CS-specific sparse training method is to attain universality. That is, the desired training method can improve the accuracy of various CS-based CNNs among different sparsities. With the training method, users do not need to customize training recipes for different CNN structures.

Conventionally, training a sparse neural network follows the process of dense training, single-shot pruning, and finetuning. We argue the conventional training process is unfavorable to CS-based CNNs under high sparsities ($> 75\%$) in which case the derived sparse masks from the dense weight tensors tend to be suboptimal. In contrast, we propose a two-phase training scheme: gradual sparse training followed by retraining. As demonstrated in Algorithm 1, assuming that the allowed training iteration amount in each phase is $I$, the first phase starts training with a dense network and the sparsity of the network discretely steps up every $P$ iterations. The completion of the first phase offers the sparse masks conforming to the pattern of CS and the weights as the initialization of the second phase. The second phase simply uses the same training setups as the first phase to retrain the retained weights in networks selected by the sparse masks. Obviously, the novelty of our training method mainly lies in the first phase. The first phase comprises two features as detailed below.

**Variable number of steps for gradual sparsification**. By empirical observation, we find the higher the required sparsity is, the more steps for gradual sparsification are needed to obtain the better model quality. Hence, to obtain a CS-based CNN at the sparsity $S$, we set $K$ steps for gradually sparsifying the network. For instances, in case that the target sparsity is 75%, the change of sparsities during training in the first phase is $0\% \to 25\% \to 50\% \to 75\%$, while for the target sparsity of 87.5%, the change is $0\% \to 12.5\% \to 25\% \to 37.5\% \to 50\% \to 62.5\% \to 75\% \to 87.5\%$. To be formulated, for the $i$th training iteration, the desired sparsity $S_i$ is:

$$S_i = \frac{ceiling\{\frac{i}{I} * K\} - 1}{K} \tag{5}$$

Equation 5 intuitively means that $S_i$ performs the piecewise linear growth as training iterations. The position of Equation 5 in the workflow of our training method is shown in Algorithm 1. Note that gradual sparsification only means the amounts of weights participating in the forward passes are gradually and discretely reduced. During backward passes, the gradients of all the weights are computed and every weight is updated no matter which sparsity step a network is at.

**CS-specific weight reselection**. Since all the weights are kept updating in the first training phase, it is beneficial to reselect the weights, i.e., update the sparse masks once in a while. Supposing every $F$ iterations, the sparse masks should be updated. The update process for CS is: 1) Reshape. A weight tensor with the shape $C_{out} \times C_{in} \times F_h \times F_w$ is reshaped into $G \times K \times L$, where $K$ and $L$ is defined in Equation 1 and 2, respectively. $G$ can be inferred given $K$ and $L$. 2) Reselection. Along $K$ axis to reselect $k_i$ weights with the highest absolute value. $k$ is related to the sparsity step that a network is at, i.e.,

$$k_i = (1 - S_i) * K \tag{6}$$

The positions of the reselected weights in masks are set ones while the other positions are set zeros. 3) Restoration, which means inversely reshape the weight tensor from $G \times K \times L$ back to $C_{out} \times C_{in} \times F_h \times F_w$. Fig. 3 shows an example of the CS-specific weight reselection procedure, which mainly visualizes the step 2). Notably, the gradual sparsification is exactly achieved by our weight reselection procedure by properly setting $F$ to make $P$ divisible by $F$.

---
**Algorithm 1** Workflow of Our Training Method

---
**Initialization:**  Dense weights $W$
**Input:**  Required sparsity $S$, training iterations $I$, data $D$
**Output:**  Sparse weights $W^S$
**Key params:**  $K$, sparse mask $M$, mask updating freq. $F$

---
1: —————————-*Gradual Sparse Training Phase*————————-
2: **for** $i = 0; i < I; i++$ **do**
3:     **if** $i\%F$ **then**
4:        $S_i \leftarrow$ Equation 5$(i, I, K)$
5:        $M \leftarrow$ Weights_Reselect( $S_i, W$ )
6:     **end if**
7:     Forward$(W, M, D)$
8:     Backward$(W, D)$
9:     Weights_Update$(W)$ #Update all weights
10: **end for**
11: ————————————*Retraining Phase*————————————
12: **for** $i = 0; i < ite; i++$ **do**
13:     Forward$(W, M, D)$
14:     Backward$(W, M, D)$
15:     Weights_Update$(W, M)$ #Update selected weights
16: **end for**
17: ———————————————————————————————
18: $W^S \leftarrow W * M$

---

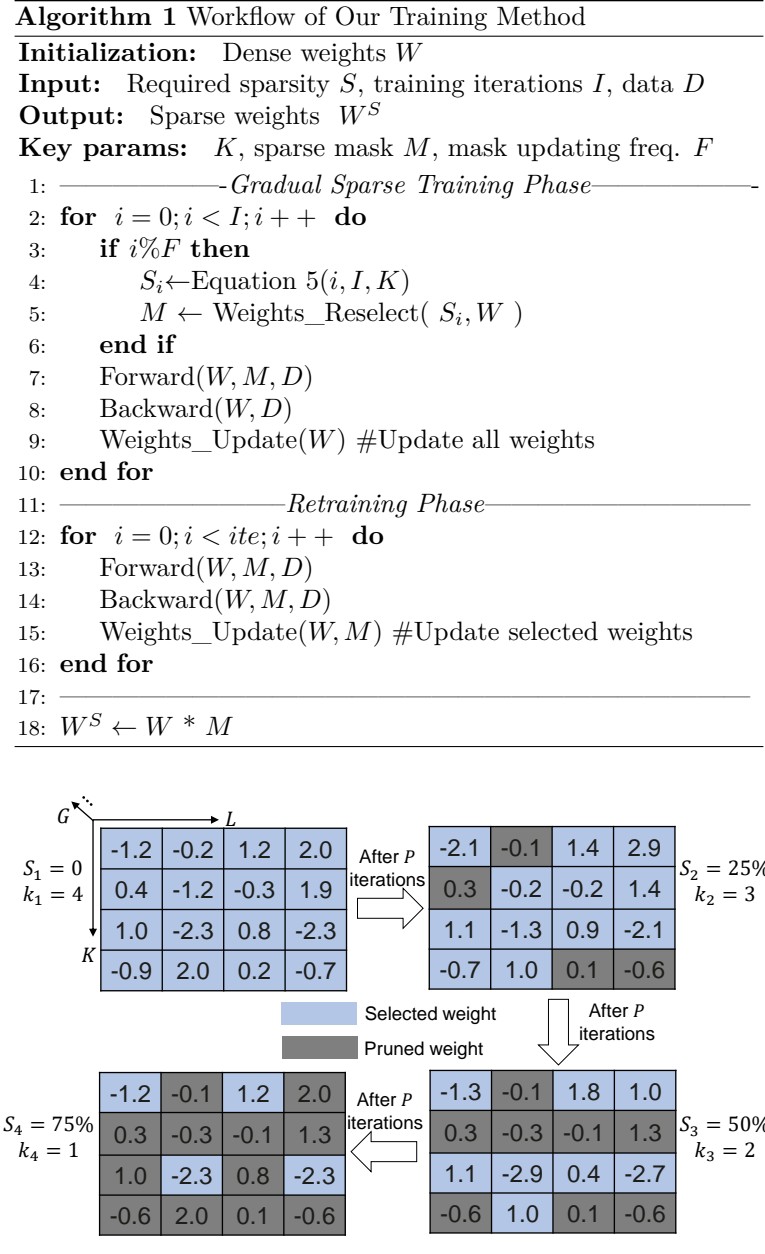

Figure 3:  An example of CS-specific weight reselection.

## 3.3  Acceleration of CS-based Convolution

The obtained sparse weight tensors after training are stored in the compressed format. That is, only the non-zero weights and their indices are stored and used for inference. This procedure is formulated by, converting $W^S$ to $W^s$ and $Idx$, where $W^s$ is the non-zero weight tensor with the smaller shapes and $Idx$ is the index tensor. The index of a non-zero weight denotes the weight's position number among $K$ complementary positions, thus the value of an index is constantly less than $K$. Fig. 4 exemplifies the compression of CS at the 50% sparsity. In this case, the $W^s$ has half of the shape than $W^S$. Although $Idx$ has the same shape as $W^s$, each index in $Idx$ can be encoded with very low numbers of bits. As shown in Table 2, only at most 4 bits are needed to encode a non-zero weight's index.

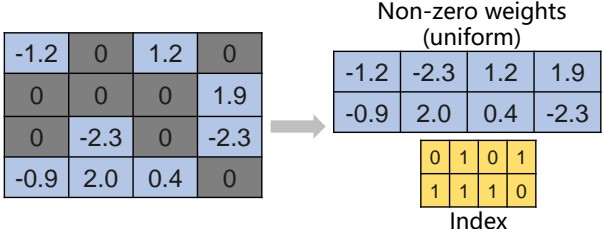

Figure 4: A diagram of CS compression at the 50% sparsity.

After acquiring $W^s$ and $Idx$, given an input activation $X$, the output featuremap $Y$ can be computed in parallel. Contrary to the conventional method of feature reuse to speedup sparse convolutions Tan et al. (2022) on common GPUs, we propose a weight reuse-based algorithm. In implicit GEMM-based dense convolutions, each thread is generally in charge of a subblock in $Y$, e.g., of size $4 * 4$. Then,

$$Y_{i:i+4,j:j+4} = \sum_{n=0}^{L} W_{i:i+4,n} \otimes X_{n,j:j+4} \tag{7}$$

In Equation 7, $L$ is only equal to $C_{in} * F_h * F_w$ for GPUs, and $\otimes$ represents the outer product operation of two vectors. When CS-based convolution is conducted, Equation 7 can be modified into:

$$Y_{i:i+4,j:j+4} = \sum_{n=0}^{L*(1-S)} W_{i:i+4,n} \circ X_{f(n,i),j:j+4}, \tag{8}$$

where

$$f(n,i) = n + M * Idx[i:i+4,n] \tag{9}$$

and $\circ$ in Equation 8 denotes the operation:

$$\circ = \begin{bmatrix} W_i * X_{n+M*Idx[i,n],j:j+4} \\ W_{i+1} * X_{n+M*Idx[i+1,n],j:j+4} \\ W_{i+2} * X_{n+M*Idx[i+2,n],j:j+4} \\ W_{i+3} * X_{n+M*Idx[i+3,n],j:j+4} \end{bmatrix} \tag{10}$$

Equation 8 makes it possible to compute CS-based convolutions by chunk. For a non-zero weight, there are $K$ related sub-blocks in an input activation that may be indexed during convolution. On common GPUs, by loading all the related $K$ sub-blocks to GPUs' shared memory in advance, Equation 8 can be conducted efficiently. Algorithm 2 shows this parallel acceleration process, where 50% CS-based convolution is taken for example.

On CPUs, the direct method is employed to accelerate CS. Due to the instruction limit, CPUs can hardly fetch multiple values far apart from each other. Accordingly, our CS allows different values of $M$ without reducing network accuracy, which is very friendly to CPU operation. During convolutions, CPUs conduct sparse vector products along the $C_{in}$ axis. In this case, $L = C_{in}/c$.

## 4 Experiments

### 4.1 Datasets, Models and Settings

CIFAR100 Krizhevsky et al. (2009) and ImageNet-1k Deng et al. (2009) are two datasets used to test the accuracy of CS-based CNNs. Specifically, on CIFAR100, we evaluate three classical CNNs including VGG-19 Simonyan & Zisserman (2014), ResNet-18 and ResNet-50 He et al. (2016), and two parameter-efficient CNNs including MobileNetv2 and SqueezeNet Sandler et al. (2018); Iandola et al. (2016). Since on CIFAR100, low sparsities ($\leq 75\%$) may result in insignificant accuracy differences between CNNs with

---
**Algorithm 2** Parallelization for 50% CS-based convolution

---
**Input:**  $Idx, W, X$
**Output:** $Y$
**Key params:**  $M, L$
___**Shared**___ **float4** $local\_w, local\_idx$
___**Shared**___ **float4** $local\_x[2], local\_y$

---
1: ————————————$Parallelism$————————————-
2: **for all**  $N * OC * OH * OW/16$  threads **do**
3:     Subblock(Y) ← SUBCONV( thread[i], $Idx, W, X$ )
4: **end for**
5: ————————————$Details$————————————
6: **function** SUBCONV(tid, $Idx, W, X$)
7:     **for**  $i = 0; i < L; i+ = BM$  **do**
8:         $local\_filter$ ← Subblock($filter$)
9:         $local\_idx$ ← Subblock($Idx$)
10:         $local\_x[0]$ ← Subblock1($X$)
11:         $local\_x[1]$ ← Subblock2($X$)
12:         Syncthreads();
13:         $local\_y =$ Eq. 8($local\_w, local\_x, local\_idx$)
14:     **end for**
15:     return $local\_y$
16: **end function**

---

our CS and with other sparse patterns, we test the three classical CNNs under high sparsities: 87.5% and 93.75%, while for MobileNetv2 and SqueezeNet, accuracy results under the 50% and 75% sparsities are adequately distinguishable for comparison. At each sparsity, each CNN is sparsified by US, filter pruning, N:M, OVW, and CS, respectively. All the sparse CNNs are firstly trained with the same training paradigm: dense training, pruning, and finetuning. This paradigm has been widely used to form various sparse CNNs. For simplicity, the finetuning phase uses the same setting as the dense training phase, which has been verified as reasonable in Mishra et al. (2021). After that, CS-based CNNs are trained with the proposed CS-specific gradual training method for comparison. All the trainings use the common and the same settings. The total training epoch is 400. For the conventional training paradigm, the dense training phase uses 200 epochs and the finetuning phase uses the rest. Similarly, for our CS-specific gradual training method, each training phase equally uses 200 epochs as well. Each experiment is repeated three times and all the experimental results on CIFAR100 are listed in the format of "mean± standard deviation" to reduce the influence of random factors.

On ImageNet-1k, CS-based ResNet-50 at different sparsities are trained for comparing with other related works. We use the officially recommended hypermeter settings for our sparse training method zlm (2022). Besides, the speedups of CS-based CNNs over the dense counterparts are measured on an Intel(R) Xeon(R) Gold 6154 CPU and a Tesla V100 GPU without sparse tensor cores, respectively.

### 4.2   Results on CIFAR100

Table 3 shows the accuracy of different networks within different sparse patterns on CIFAR100. Firstly, for the three classical CNNs, our CS achieves the best accuracy under high sparsities among all the sparse patterns. Secondly, for MobileNetv2 and SqueezeNet, our CS also outperforms other fine-grained structured sparse pattern at the 75% sparsity. In particular, the accuracy of MobileNetv2 within CS at the 75% sparsity is even higher than that within the unstructured sparsity, i.e., 62.63%>61.7%. Thirdly, the proposed training method significantly improves the accuracy of a series of CS-based CNNs, with the average accuracy increasing from 0.48% to 2.83%. Notably, at the 50% sparsity, all types of sparse patterns lead to lossless accuracy. In this case, we argue the accuracy differences among the patterns and training methods are immaterial.

| Spa. (%) | | VGG19 | Resnet18 | Resnet50 | Spa. (%) | | Mobilenetv2 | SqueezeNet |
|---|---|---|---|---|---|---|---|---|
| | Origin | 70.8 | 74.7 | 72.2 | | Origin | 65.3 | 65.1 |
| 87.5 | Unstructured | 71.3±0.1 | 73.3±0.1 | 72.3±0.3 | 50 | Unstructured | 66.9±0.1 | 67.7±0.2 |
| | Filter pruning | 47.8±0.6 | 59.0±0.2 | 48.4±1.0 | | Filter pruning | 65.5±0.5 | 36.4±0.2 |
| | N:M(1:8) | 70.6±0.1 | 72.9±0.1 | 72.1±0.1 | | N:M(2:4) | 66.4±0.3 | 67.2±0.1 |
| | OVW | 63.6±0.3 | 64.1±0.7 | 59.2±1.6 | | OVW | 61.8±0.3 | 64.0±0.4 |
| | **CS-C** | 70.7±0.2 | 73.0±0.1 | 72.5±0.3 | | **CS-C** | 66.4±0.1 | 67.2±0.1 |
| | **CS (Ours)** | **71.7±0.3** | **73.5±0.1** | **73.1±0.0** | | **CS (Ours)** | **66.7±0.2** | **67.0±0.2** |
| | △ | +1.4 | +0.5 | +0.6 | | △ | +0.2 | -0.2 |
| 93.75 | Unstructured | 69.8±0.1 | 71.5±0.0 | 71.3±0.0 | 75 | Unstructured | 61.7±0.1 | 64.5±0.2 |
| | Filter pruning | 33.2±0.4 | 47.9±0.4 | 40.0±0.6 | | Filter pruning | 53.4±1.1 | 15.9±0.2 |
| | N:M(1:16) | 68.1±0.3 | 70.4±0.2 | 70.6±0.3 | | N:M(1:8) | 58.9±0.4 | 63.4±0.3 |
| | OVW | 59.6±0.2 | 60.6±0.3 | 41.0±13.8 | | OVW | Failed | 54.6±0.9 |
| | **CS-C** | 68.5±0.2 | 70.5±0.1 | 70.5±0.2 | | **CS-C** | 59.8±0.1 | 63.9±0.3 |
| | **CS (Ours)** | **69.9±0.1** | **72.2±0.3** | **73.0±0.3** | | **CS (Ours)** | **62.6±0.2** | **64.4±0.1** |
| | △ | +1.4 | +1.7 | +2.5 | | △ | +2.8 | +0.5 |

Table 3: Accuracy of sparse CNNs on CIFAR100. 'Spa.' means sparsity. 'CS-C' represents **CS**-based CNNs formed by the **C**onventional training paradigm, while 'CS (Ours)' means that formed by the proposed training method. '△' means the difference between 'CS-C' and 'CS (Ours)'. For spatial brevity, all the data are rounded to one significant digit.

Apart from Table 3, the experimental results on CIFAR100 using the same network for all sparsity ratios are shown in Figure 5. It is observed that the curves of our CS are generally above the curves of other structured patterns and are overlapped with curves of unstructured sparsity. That observation further demonstrates the superiority of our CS in terms of accuracy across all the sparsity ratios.

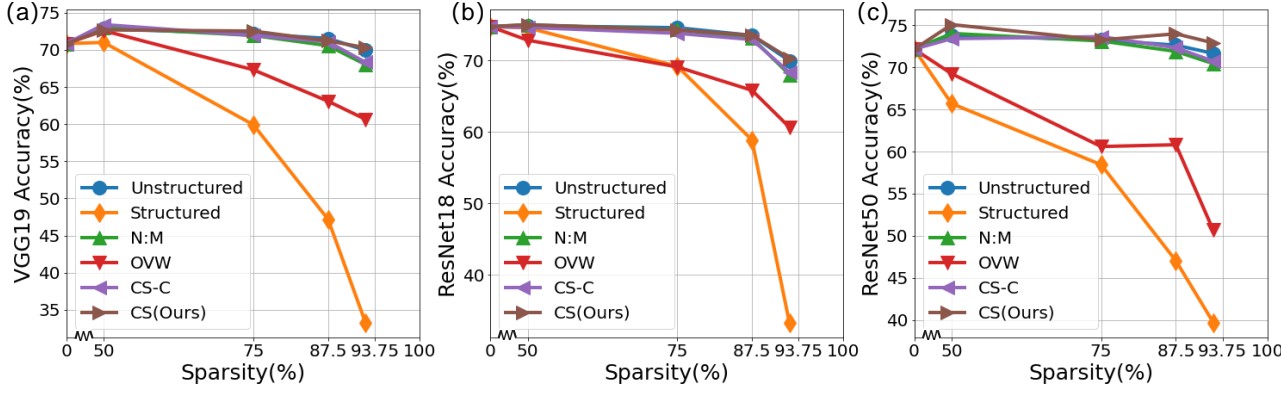

Figure 5: Accuracy vs. sparsity curves of different CNNs whitin different sparse patterns. (a) VGG19. (b) ResNet18. (c) ResNet50.

Notably, on CIFAR100, our CS performs even better than unstructured sparsity due to the relatively small dataset size. Generally, compared to modern DNNs' capacity, CIFAR100 is easy to be overfitted. Thus, within a certain sparsity range, weight sparsification can largely regularize the model capacity and lead to higher accuracy over dense counterparts (Hoefler et al., 2021). Compared to unstructured sparsity, fine-grained structured sparsity such as CS and N:M constrains a model more strictly. This constraint generally endows the model favorable regularization on small datasets like CIFAR100 and at low sparsity like 50% and 75%. For example, as shown in Figure 7, at 50% sparsity for VGG19 on CIFAR100, N:M sparsity leads to 73.00% accuracy, which is better than 72.82% of unstructured sparsity. At 75% sparsity, our CS-based VGG19 reaches 72.51% accuracy, while unstructured sparsity-based VGG19 is 72.13%.

Under higher sparsity such as 93.75%, despite the rare possibility of overfitting, the accuracy gap between fine-grained structured patterns like 'CS-C' and 'unstructured' was still narrow due to the small dataset size. Hence, with our proposed training scheme, 'CS(Ours)' can easily fill the gap to further become on par with or to even surpass 'unstructured'. By contrast, on ImageNet, our CS and other fine-grained sparsity patterns bring constantly lower accuracy than unstructured sparsity regardless of sparsity ratios and network architectures.

### 4.3 Results on ImageNet

Table 4 shows the experimental results on ImageNet. Compared with the OVW and Shfl_BW patterns, our CS with the proposed training scheme leads to better accuracy under high sparsities, e.g., 93.75%. For other sparsities, our CS achieves comparable accuracy with the state-of-the-art N:M sparsity. However, the different settings of $M$ in N:M sparsity significantly affect the network accuracy, e.g., Sun et al. (2021). On the contrary, our CS is robust to the pattern hyperparameter setting which will be shown in the ablation study.

| Pattern | Sparsity | Error(%) | | | Params | Flops |
|---|---|---|---|---|---|---|
| | | Ori. | Pruned | Gap | (M) | (G) |
| N:M | 2:4 | 77.3 | 77.0 | 0.3 | 13.8 | 2.15 |
| OVW | 50% | 76.12 | 75.76 | 0.36 | 13.8 | 2.15 |
| **CS(Ours)** | **50%** | **76.39** | **76.37** | **0.02** | **13.8** | **2.15** |
| N:M | 1:4 | 77.3 | 75.3 | 2 | 7.93 | 1.17 |
| OVW | 70% | 76.12 | 73.35 | 2.77 | 9.14 | 1.37 |
| **CS(Ours)** | **75%** | **76.39** | **75.18** | **1.21** | **7.93** | **1.17** |
| Shfl_BW | 80% | N/A | 75.94 | N/A | 6.78 | 0.98 |
| **CS(Ours)** | **87.5%** | **76.39** | **72.44** | **3.95** | **5.02** | **0.69** |
| N:M | 1:16 | 77.3 | 71.5 | 5.8 | 3.52 | 0.44 |
| Shfl_BW | 90% | N/A | 73.09 | N/A | 4.43 | 0.59 |
| **CS(Ours)** | **93.75%** | **76.39** | **71.07** | **5.32** | **3.52** | **0.44** |

Table 4: ResNet50 accuracy comparison among different fine-grained structured sparse patterns on ImageNet. The results of N:M, OVW, and Shfl_BW are from Zhou et al. (2021a), Tan et al. (2022) and Huang et al. (2022), respectively. 'N/A' means the related work does not report the result.

Besides, although this work mainly focuses on CNNs, we also give the preliminary results of CS and N:M sparsity on Transformer structures as shown in Table 5. The DeiT-small is used and the training settings of CS-based DeiT-small and N:M-based DeiT-small are identical. That is, our proposed training scheme is not used this time for an absolutely fair comparison. Experimental results show once more that our CS also incurs a comparable accuracy as N:M sparsity on Transformer architectures. Note that at 50% sparsity, both N:M sparsity and CS -based DeiT-small surpass the dense one, so the difference between the two sparse patterns at 50% sparsity is trivial.

With the experimental results of CNN and Transformer architectures, we would like to reaffirm that our CS mainly achieves comparable accuracy to N:M sparsity under identical training settings. The role of our proposed training scheme is only to improve a CS-based network's accuracy under high sparsity ratios. For instance, at 93.75%, ResNet50 within CS surpasses that within N:M sparsity by a decent margin, i.e., ~0.5% on ImageNet as shown before in Table 4. In other words, a better acceleration affinity on CPUs and common GPUs under comparable accuracy, is truly our CS's advantage over N:M sparsity.

Finally, due to the slow progress in the explainability of DNNs, the model accuracy of CS lacks clear theoretical support. However, we found a metric called mask diversity that may provide some insights (Hubara et al., 2021). The metric is defined as the number of possible masks for a sparse pattern given a sparsity ratio. For example, for a $8 \times 8$ weight tensor and a 50% sparsity, our CS has $2^{32} = 4,294,967,296$ possible masks, while other vector-wise sparsity such as OVW can only have $\binom{16}{8} = 12870$ masks, not to

mention only $\binom{8}{4} = 70$ masks that channel pruning can provide. Altogether, CS's high mask flexibility probably promotes the high accuracy of CS-based CNNs.

| Sparsity(%) | 50 | 75 | 87.5 | 93.75 |
|---|---|---|---|---|
| N:M | 77.61 | 73.69 | 67.56 | 60.08 |
| CS(Ours) | 77.4 | 73.66 | 67.63 | 60.01 |

Table 5: Comparison between CS and N:M sparsity using Top1 accuracy of DeiT-small on ImageNet. The dense DeiT-small is 75.56%.

## 4.4 Ablation Study

Firstly, we investigate the effect of different mask updating frequencies in our training method, i.e., $F$ mentioned in Algorithm 1, on network accuracy. The results are shown in Table 6. In the table, $F = 0$ means updating the mask once for every iteration, $F = 0.5$ means that updating frequency is 0.5 epoch, and so on. We find that the higher the required sparsity is, the higher the mask updating frequency should be. For example, at the 50% sparsity, $F = 8$ is the best, while at 93.75%, $F = 0$ outperforms others. The $F$ settings in Table 6 is exactly used in training CS-based ResNet50 on ImageNet. Secondly, we investigate a pattern hypermeter of CS: $M$. Specifically, under the same sparsity decided by $K$, CS-based CNNs with different settings of $M$ are trained and we conduct pairwise t-test on these CNNs' accuracy. As shown in Table 7, all the $p$ values are larger than 0.05, which indicates that our CS is robust to pattern hyperparameters. The robustness is quite beneficial for acceleration as CPUs and GPUs can employ different values of $M$ to meet the respective constraints in memory access and instruction set.

| $F$ | 50% | 75% | 87.5% | 93.75% |
|---|---|---|---|---|
| 0 | 74.7±0.08 | 74.14±0.63 | 73.44±0.19 | **72.06±0.38** |
| 0.5 | 74.78±0.09 | **74.55±0.06** | **73.57±0.27** | 71.9±0.52 |
| 1 | 74.68±0.06 | 74.35±0.37 | 73.47±0.6 | 71.62±0.2 |
| 2 | 74.71±0.24 | 74.12±0.23 | 73.16±0.37 | 71.19±0.59 |
| 4 | 74.85±0.08 | 74.21±0.08 | 72.85±0.23 | 70.85±0.48 |
| 8 | **74.93±0.37** | 74.28±0.21 | 72.54±0.61 | 70.47±0.17 |
| 10 | 74.66±0.19 | 74.5±0.19 | 73.06±0.17 | 70.14±0.09 |

Table 6: ResNet18 on CIFAR100: Impact of $F$ across sparsities.

| Sparsity | $(M,K)$ | VGG19 | ResNet18 | ResNet50 | Sparsity | $(M,K)$ | MobileNetv2 | SqueezeNet |
|---|---|---|---|---|---|---|---|---|
| 87.5% | (2,8) | 71.28±0.15 | 73.68±0.32 | 73.46±0.85 | 50% | (2,2) | 66.81±0.14 | 67.19±0.29 |
| | (4,8) | 71.46±0.49 | 73.2±0.39 | 73.23±0.65 | | (4,2) | 66.65±0.15 | 67.07±0.04 |
| | (8,8) | 71.55±0.39 | 73.47±0.16 | 73.38±0.39 | | (8,2) | N/A | 67.27±0.09 |
| | **t-test** | **0.77** | **0.29** | **0.82** | | **t-test** | **0.18** | **0.57** |
| 93.75% | (2,16) | 69.62±0.09 | 71.89±0.19 | 72.23±1.4 | 75% | (2,4) | 62.72±0.14 | 64.26±0.07 |
| | (4,16) | 69.76±0.19 | 72.22±0.39 | 72.56±0.11 | | (4,4) | N/A | 64.3±0.22 |
| | **t-test** | **0.31** | **0.39** | **0.71** | | **t-test** | N/A | **0.7** |

Table 7: Accuracy of CS-based CNNs with different settings of $M$ on CIFAR100

## 4.5 Results on Speedups

On CPU, $M$ is set 2,4,8,16 on demand. On GPUs, we set $M$ as large as possible, i.e., $M = C_{in} * F_h * F_w / K$. We find that the larger $M$ makes indexing more easier on GPU. Fig. 6 shows the normalized speedups on CS. Firstly, three typical convolutions in ResNet50 are used for test speedup. As shown in Fig. 6(a), with batch size equal to 1, CPU achieves 4.27×, 5.46× and 7.7× speedups for the 3rd, 11th, 41th convolutions at the

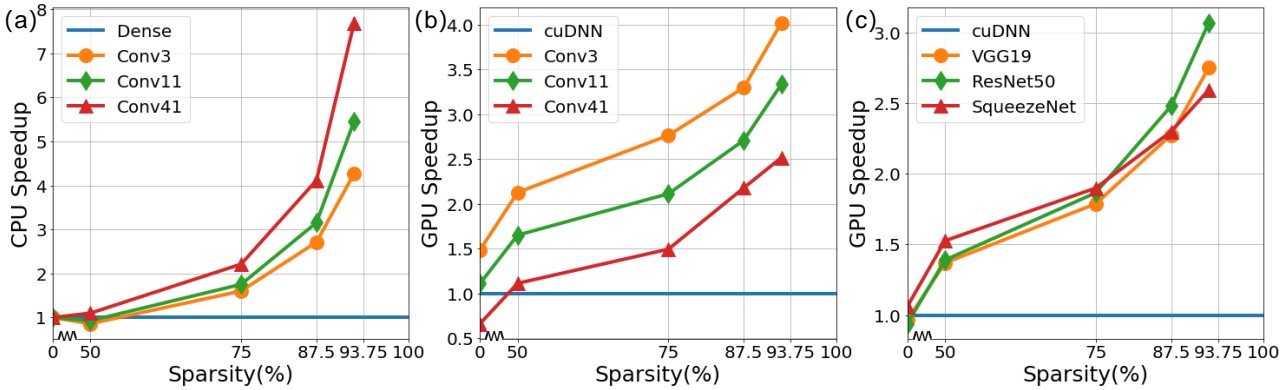

Figure 6: Speedups for CS. (a) CPU speedups for three CS-based convolutions in ResNet50. (b) GPU speedups for three CS-based convolutions in ResNet50. (c) GPU speedups for three CS-based CNNs.

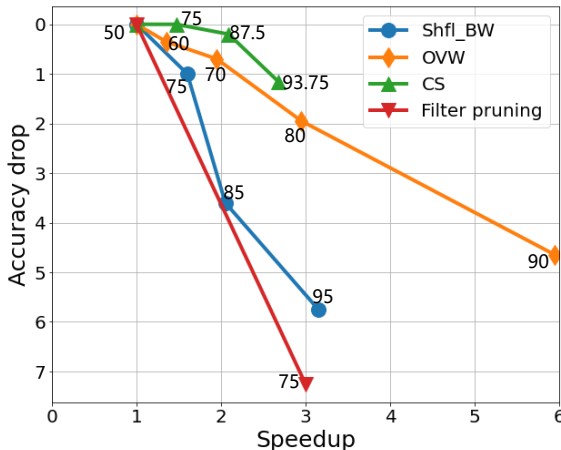

Figure 7: ResNet50 on CIFAR100: Normalized accuracy-speedup curves of three fine-grained structured sparse patterns. Note N:M sparsity does not have acceleration gains on common GPUs.

93.75% complementary sparsity, respectively. Similarly, as shown in Fig. 6(b), with batch size equal to 64, GPU respectively achieves $4.02\times$, $3.33\times$ and $2.52\times$ speedups at 93.75% over dense counterparts supported by cuDNN. Secondly, we also estimate the speedups on network-level by averaging all the convolutions' runtime. GPU achieves $2.75\times$, $3.07\times$, and $2.59\times$ speedups for VGG19, ResNet50 and SqueezeNet, respectively. These speedup performances prove the efficiency of the proposed parallel acceleration algorithm. In addition, our CS generally reaches better accuracy-speedup tradeoffs compared with the OVW and Shfl_BW pattern, as shown in Fig. 7.

## 4.6 Results on Practical Inference Performance

Table 8 shows the speedup comparison of CS and N:M sparsity on A100 and V100 GPUs, respectively. Note that the speedup data of N:M sparsity are directly cited from A100 materials that are publically available. Although A100 GPUs have a far stronger computing power than previous ones, it is noteworthy that A100 GPUs are quite inflexible as it can only support the acceleration of 2:4 sparsity. Other sparsity ratios, such as 75% (1:4), 87.5% (1:8), and 93.75% (1:16) **can not** incur any speedups on A100. In contrast, our CS supports a wide range of sparsity ratios on common GPUs.

|  | 50% | 75% | 87.5% | 93.75% |
|---|---|---|---|---|
| N:M (A100) | 2 | 0 | 0 | 0 |
| CS (V100) | 1.39 | 1.86 | 2.48 | 3.07 |

Table 8: ResNet50 speedups comparison at different sparsities on V100 and A100

In addition, CS-based ResNet50 inference energy consumptions at different sparsities are shown in Table 9. As the sparsity ratio arises, the reduced FLOPs efficiently convert to the shorter inference time. At the 93.75% sparsity and 64 batch size, CS-based ResNet50 only consumes 3.28J energy for one inference, which is energy-efficient for cloud server scenarios.

| Sparsity(%) | Gflops | Times(ms) | Energy cost(J) |
|---|---|---|---|
| 50 | 2.15 | 28.98 | 7.25 |
| 75 | 1.17 | 21.59 | 5.4 |
| 87.5 | 0.69 | 16.24 | 4.06 |
| 93.75 | 0.44 | 13.12 | 3.28 |

Table 9: Energy consumption of ResNet50 within CS pattern. $batch\_size = 64$

## 5 Conclusion and Future Work

We propose a novel CS to accelerate sparse CNNs on CPUs and common GPUs and retain the network accuracy as much as possible. To our knowledge, we are the first to report the practical speedups on both CPUs and common GPUs for a sparse pattern. Not only does the proposed CS feature high mask flexibility that contributes a lot to sparse CNNs' accuracy, but also the network accuracy is robust to pattern hyperparameters. The robustness enhances CS's adaptability to different computing platforms.

we would like to denote that the limitation of our proposed training scheme for CS is that the scheme is hardly viable for massive pretraining data. It is because our training scheme needs to be embedded into the pretraining stage and huge amounts of pretraining data make it actually impossible to replicate the pretraining. Thus, it is difficult for our scheme to be applied in scenarios such as LLMs. However, we argue that the limitation does not disqualify our major contributions for CS and we will handle the limitation in future work.

## 6 Acknowledgement

We gratefully acknowledge the support of MindSpore (Hua, 2020), CANN (Compute Architecture for Neural Networks) and Ascend AI Processor used for this research.

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
