# OpenReview forum: "Complementary Sparsity: Accelerating Sparse CNNs with High Accuracy on General-Purpose Computing Platforms"
_TMLR — Accepted by TMLR_

### Review · Reviewer_SjDi · 2023-08-30

**Summary Of Contributions:**

This paper presents a new fine-grained pruning method termed "complementary sparsity". Similar to the more widely-known N:M sparsity, this paper presents a 1:K strided sparsity pattern in CNNs. This means that for a strided group of weights along the channel dimension, every group of weights that are K elements apart, only one can have a value while the rest are set to zero. This feels like a specific case of N:M sparsity, but the authors of the paper point out that an efficient GPU kernel can be written to accelerate the proposed complementary sparsity, while N:M sparsity requires special sparsity support in Ampere GPUs.

**Audience:**

Yes

**Claims And Evidence:**

No

**Requested Changes:**

Addressing the weaknesses above would make the paper much better in my opinion. Specifically:
- More experiments and demonstrating generality to different models, tasks, datasets.
- A clarification of the training regime and ensuring a fair comparison.
- Comparison to N:M sparsity from a perf/watt perspective.

**Strengths And Weaknesses:**

Strengths:
- Fine-grained unstructured sparsity us plentiful in DNNs yet challenging to accelerate, and the proposed method moves towards finer-grained semi-structured sparsity that can be practically accelerated on commodity GPUs.
- Both a sparsity algorithm is presented and real hardware speedups are realized through a custom GPU kernel.

Weaknesses:
- Evaluation is very limited to five somewhat dated CNNs. Evaluations on other DNN topologies (e.g. Transformers) and other tasks (e.g. speech, language, generative) would better demonstrate the efficacy of a new supposedly-general pruning approach.
- I have some concerns about whether the accuracy comparisons are fair. A special training method was applied for the proposed complementary sparsity method. What about all the baselines? How were they trained? Hyperparameters, epochs, etc? Do you use the same gradual pruning approach for the baselines?
- Improvements vs. baselines are somewhat limited.
- No comparisons to N:M sparsity for which the support exists on current GPUs. I realize that authors wanted to present a method without dedicated/additional GPU support, but a comparison would've been informative.
- [Minor] No energy/power measurements. Would be interesting to see if there are energy implications.
- Are you releasing your code?

---

> ### Author Response · Authors · 2023-09-16
> **Some experimental results and explanations-1**
>
> Thank you very much for reviewing our manuscript.
>
> Q1: I have some concerns about whether the accuracy comparisons are fair. A special training method was applied for the proposed complementary sparsity method. What about all the baselines? How were they trained? Hyperparameters, epochs, etc? Do you use the same gradual pruning approach for the baselines?
>
> A1: Thanks for your concern. 1) In the experiments section of our manuscripts, the accuracy of sparse CNNs on CIFAR100, i.e., Table 3, is obtained with the same training hyperparameters and epochs. In particular, we have presented in Table 3 the results of our CS under identical training settings with other sparse patterns, e.g., '**CS-C**' rows. By comparison, '**CS(Ours)**' rows are results using our proposed training scheme.
>  2) Accuracy results on ImageNet of ResNet50 with N:M, OVW, and Shfl\_BW patterns in Table 4 are directly cited from related works. Replicating the works of N:M, OVW, and Shfl\_BW patterns on ImageNet with multiple sparsity ratios is prohibitively time-consuming. In a sense, it is very hard to ensure an absolutely fair comparison on the ImageNet dataset. Nevertheless, we list the training settings of our work and related works on ImageNet as shown below for your information.
> | Hyperparameters    |          Shfl_BW          |       OVW         |              N:M                 | Ours   |
> |:------------------:|:----------------------:|:-----------------:|:--------------------------------:|:--------------------------:|
> |       Scheme       |     Pretrain+finetune  | Pretrain+finetune |       Training from scratch      |    Pretrain+finetune       |
> |    Batch_size      |         N/A            |       N/A         |              256                 |            32              |
> |      Epochs        |         N/A            |       100         |              120                 |            90              |
> |    Initial lr      |         N/A            |   0.1/0.0008      |              0.1                 |           0.1              |
> |     Momentum       |         N/A            |       N/A         |              N/A                 |           0.9              |
> |    Extra method    | Heuristic searching    | Row clustering    | Sparse refined regularization    |     Gradual pruning        |
>
> Table 7. Training settings on ImageNet. Note for 'Pretrain+finetune' schemes, overall epochs should be doubled.
>
>  Note that every work applies an extra method beyond the naive training scheme to improve their model accuracy. Besides, although the state-of-the-art N:M work only employs 120 epochs in their training scheme, the time of one epoch in their scheme is much longer than that in other works' training schemes. It is due to their sparse-refined regularization that incurs extra computations in each iteration of training.
>
> Q2: Improvements vs. baselines are somewhat limited.
>
> A2: Thanks for your comment. We also claim in our manuscript that our complementary sparsity (CS) mainly achieves a **comparable** network accuracy with N:M sparsity at most of the sparsity ratios. Note that under high sparsity such as 93.75\%, sparse CNNs' accuracy within CS can surpass that within N:M sparsity by a decent margin, i.e., \textasciitilde 0.5\% on ImageNet as shown in Table 4. This margin is exactly caused by the proposed training scheme. Besides, we argue that our CS is not completely a special case of N:M sparsity as our CS is robust to hyperparameters while N:M sparsity is not. For example, under the same sparsity like 87.5\%, CNNs within 2:16 sparsity usually have significantly better accuracy than those within 1:8 sparsity. In contrast, CNNs within CS of $(2,8), (4,8)$, and $(8,8)$ settings in terms of $(M, K)$ have the similar accuracy as shown in Table 6. That property enables us to adjust the hyperparameters of CS according to different hardware platforms for better acceleration performance. However, your views are highly valued and we will continue to study the relation between N:M and CS theoretically in the near future.
>
> We prefer to highlight the hardware acceleration generality of our CS. Although our CS achieves comparable or moderately better accuracy over N:M sparsity, CS can promote significant speedups on common GPUs and CPUs while N:M sparsity is explicitly weak in this aspect.

---

> ### Author Response · Authors · 2023-09-16
> **Some experimental results and explanations-2**
>
> Q3: No comparisons to N:M sparsity for which the support exists on current GPUs. I realize that the authors wanted to present a method without dedicated/additional GPU support, but a comparison would've been informative.
>
> A3: Thank you very much for your understanding. Due to the sanctions, we are not accessible to A100 which has sparse tensor cores to support 2:4 sparsity. Nevertheless, we carefully refer to the A100 materials publicly available for some informative comparisons with our CS. Table 8  shows some results. We would like to stress the inflexibility of A100 which means that it can only support the acceleration of 2:4 sparsity. Other sparsity ratios, such as 75\% (1:4), 87.5\% (1:8), and 93.75\% (1:16) can not incur any speedups on A100. In contrast, our CS supports a wide range of sparsity ratios on common GPUs.
> |        Sparsity(%)       |  50 |  75 | 87.5 | 93.75 |
> |:-------------:|:----:|:----:|:------:|:------:|
> |   N:M (A100)  |   2  |   0  |    0   |    0   |
> | **CS (V100)** | 1.39 | 1.86 |  2.48  |  3.07  |
>
> Table 8. ResNet50 speedups comparison at different sparsities on V100 and A100
>
> Q4: [Minor] No energy/power measurements. Would be interesting to see if there are energy implications.
>
> A4: Thanks. We present CS-based ResNet50 inference energy consumptions at different sparsities as shown in Table 9. As the sparsity arises, the reduced Gflops efficiently convert to the shorter inference time. At the 93.75\% sparsity and 64 batch size, CS-based ResNet50 only consumes 3.28J energy for one inference, which is energy-efficient for cloud server scenarios.
> | Sparsity(%) | Gflops | Times(ms) | Energy cost(J) |
> |:-----------:|:------:|:---------:|:--------------:|
> |      50     |  2.15  |   28.98   |      7.25      |
> |      75     |  1.17  |   21.59   |       5.4      |
> |     87.5    |  0.69  |   16.24   |      4.06      |
> |    93.75    |  0.44  |   13.12   |      3.28      |
>
> Table 9. Energy consumption of ResNet50 within CS pattern. $batch\\_size=64$
>
> Q5: Are you releasing your code?
>
> A5: Yes, we will release our code once our paper is accepted.
>
> Q6: Evaluation is very limited to five somewhat dated CNNs. Evaluations on other DNN topologies (e.g. Transformers) and other tasks (e.g. speech, language, generative) would better demonstrate the efficacy of a new supposedly-general pruning approach.
>
> A6: We give the preliminary results of CS and N:M sparsity on Transformer structures as shown in Table 10. The DeiT-small is used and the training settings of CS-based DeiT-small and N:M-based DeiT-small are identical. That is, our proposed training scheme is not used this time for your concern. Experimental results show once more that our CS incurs a comparable DNN accuracy as N:M sparsity. Note that at 50\% sparsity, both N:M sparsity and CS -based DeiT-small surpass the dense one, so the difference between the two sparse patterns at 50\% sparsity is trivial.
>
> | Sparsity(%) |   50  |   75  |  87.5 | 93.75 |
> |-------------|:-----:|:-----:|:-----:|:-----:|
> |     N:M     | 77.61 | 73.69 | 67.56 | 60.08 |
> |   CS(Ours)  | 77.4 | 73.66 | 67.63 | 60.01 |
>
> Table 10. Comparison between CS and N:M sparsity
> using Top1 accuracy of DeiT-small on ImageNet. The dense DeiT-small is 75.56\%.
>
> Besides, we honestly denote that the limitation of our proposed training scheme for CS is that the scheme is hardly viable for massive pretraining data. It is because our training scheme needs to be embedded into the pretraining stage and huge amounts of pretraining data make it actually impossible to replicate the pretraining. Thus, it is difficult for our scheme to be applied in scenarios such as LLMs and some generative tasks. We will handle the limitation in our future work.
>
> All the changes have been included in the main texts and Appendices of the newly uploaded version. It is our pleasure to respond to your valuable comments. We are very grateful for your time and patience.

---

### Review · Reviewer_r32s · 2023-09-03

**Summary Of Contributions:**

1. The paper introduces a novel sparse pattern called CS, optimized for high mask flexibility and computational acceleration, allowing for effective pruning of CNNs with minimal loss in accuracy.
2. It proposes a CS-specific training algorithm that enhances the accuracy of highly sparse CNNs, outperforming traditional N:M sparsity methods, such as 0.48% improvements on ResNet50.
3. It proposes a parallel acceleration algorithm for the inference phase, tailored for CS-based convolutions, achieving up to 3.07x speedup over dense configurations supported by cuDNN at 93.75% sparsity.

**Audience:**

Yes

**Broader Impact Concerns:**

No concerns.

**Claims And Evidence:**

Yes

**Requested Changes:**

1. page 7 we propose a weight resue-based => reuse-based
2. Please also discuss the training overhead of the CS compared with other baseline pruning methods.
3. Add performance of channel pruning and block sparsity to the Figure 6.
4. Although not all GPU supports 2:4 sparsity, it would still be valuable to compare the speed of proposed method with N:M sparsity on GPU with N:M tensor core
5. Please address the concern in weakness above.
6. fixed the overlapped "Sparsity" with column line in table 3
7. table 3, use the same network for all sparsities, so that the insight about trend of accuracy improvements can be obtained.
8. it is surprising that the CS can outperform unstructured sparsity by such a large amount. More explanations and insights are needed.

**Strengths And Weaknesses:**

strength:
1. The complementary sparsity is an interesting new idea of achieving sparsity.
2. Comparisons on CIFAR and Imagenet shows better speedup and accuracy.

Weakness
1. It seems that the propose CS requires specific optimization of the GPU kernel. It is thus unclear that when other sparsity patterns are optimized on cuda kernel level on a common GPU, whether the proposed CS sparsity can have better speed.

---

> ### Author Response · Authors · 2023-09-17
> **Reply to your comments-1**
>
> Q1: Page 7 we propose a weight resue-based => reuse-based
>
> A1: Thanks for your carefulness. We corrected the word and revised all the typos in the manuscript.
>
> Q2: Please also discuss the training overhead of the CS compared with other baseline pruning methods.
>
> A2: We compare CS-based CNNs with other types of sparse CNNs on CIFAR100 and ImageNet, respectively. On CIFAR100, all the trainings use the same hyperparameters and epochs. In particular, we have presented in Table 3 in our manuscript the results of our CS under identical training overheads with other sparse patterns, e.g., '**CS-C**' rows, for an absolutely fair comparison. Moreover, '**CS(Ours)**' rows are results using our proposed training scheme. Some of the results are posted below for your reading convenience:
> | Sparsity(%) | 　           | VGG19    | Resnet18 | Resnet50 |
> |-------------|--------------|----------|----------|----------|
> |             | **CS-C**     | 70.7±0.2 | 73.0±0.1 | 72.5±0.3 |
> |     87.5    | **CS(Ours)** | 71.7±0.3 | 73.5±0.1 | 73.1±0.0 |
> |             | △            | 1.4      | 0.5      | 0.6      |
> |             | **CS-C**     | 68.5±0.2 | 70.5±0.1 | 70.5±0.2 |
> |    93.75    | **CS(Ours)** | 69.9±0.1 | 72.2±0.3 | 73.0±0.3 |
> |             | △            | 1.4      | 1.7      | 2.5      |
>
> On ImageNet, accuracy results of ResNet50 with N:M, OVW, and Shfl\_BW patterns are directly cited from related works. Replicating the works of N:M, OVW, and Shfl\_BW patterns on ImageNet with multiple sparsity ratios is prohibitively time-consuming. In a sense, it is very hard to ensure an absolutely fair comparison on the ImageNet dataset. Nevertheless, we list the training settings of our work and related works on ImageNet as shown below for your information.
>
> | Hyperparameters    |          Shfl_BW          |       OVW         |              N:M                 | Ours   |
> |:------------------:|:----------------------:|:-----------------:|:--------------------------------:|:--------------------------:|
> |       Scheme       |     Pretrain+finetune  | Pretrain+finetune |       Training from scratch      |    Pretrain+finetune       |
> |    Batch_size      |         N/A            |       N/A         |              256                 |            32              |
> |      Epochs        |         N/A            |       100         |              120                 |            90              |
> |    Initial lr      |         N/A            |   0.1/0.0008      |              0.1                 |           0.1              |
> |     Momentum       |         N/A            |       N/A         |              N/A                 |           0.9              |
> |    Extra method    | Heuristic searching    | Row clustering    | Sparse refined regularization    |     Gradual pruning        |
>
> Table 7. Training settings on ImageNet. Note for 'Pretrain+finetune' schemes, overall epochs should be doubled.
>
> Notably, the proposed training scheme theoretically brings down the total training float-point operations (FLOPs) as we do not constantly maintain a dense network during pretraining. In practice, we found the reduction in FLOPs does not significantly shorten the total training time as the weight re-selection and scheduling the sparsity of sparse CNNs in our scheme also incur overheads. This type of overheads is generally not suitable to be measured with the FLOPs metric. On the whole, trainings with and without our scheme the under the same hyperparameters, epochs, and hardware take the closely similar time. For example, training a N:M-based CNN on ImageNet takes **65.37** hours, while for a CS-based CNN the time is **65.23** hours.

---

> ### Author Response · Authors · 2023-09-17
> **Reply to your comments-2**
>
> Q3: Add performance of channel pruning and block sparsity to the Figure 6.
>
> A3: Thank you for your remainder. The performance of channel pruning is nowadays regarded as completely equivalent to that of filter pruning. This is becasue pruning filters in a layer is exactly equal to pruning the corresponding channels in the next layer. As for block sparsity, we found that the accuracy of ResNet50 within block sparsity is collapsed (<5\%) under high sparsity ratios, e.g., 75\%, 87.5\%, and 93.75\%, which is unsuitable for being integrated into Figure 6. However, we below pesent the runtime and speedups of block sparsed ResNet50 for your information:
> | Sparsity(%) | Cutlass-based version |         | Original version |         |
> |:-----------:|:---------------------:|:-------:|:----------------:|:-------:|
> |             |        Time(ms)       | Speedup |     Time(ms)     | Speedup |
> |      0      |          2.21         |    1    |       3.76       |    1    |
> |      50     |          1.91         |   1.16  |       2.79       |   1.35  |
> |      75     |          1.98         |   1.12  |       2.07       |   1.82  |
> |     87.5    |          1.91         |   1.16  |       1.53       |   2.46  |
> |    93.75    |          1.9          |   1.16  |       0.98       |   3.84  |
>
> Table 11. The speedups of block sparsity-based ResNet50
>
> Note that GPU acceleration of block sparsity generally has two versions, the original and the Cutlass-based versions. Both versions' speedup performances are inferior to ours.  For more information about Culass, please see the newly uploaded Appendices.
>
> Q4: Although not all GPU supports 2:4 sparsity, it would still be valuable to compare the speed of proposed method with N:M sparsity on GPU with N:M tensor core.
>
> A4: Thanks. Table 8  shows some results. We would like to stress the inflexibility of A100 which means that it can only support the acceleration of 2:4 sparsity. Other sparsity ratios, such as 75\% (1:4), 87.5\% (1:8), and 93.75\% (1:16) can not incur any speedups on A100. In contrast, our CS supports a wide range of sparsity ratios on common GPUs.
> |        Sparsity(%)       |  50 |  75 | 87.5 | 93.75 |
> |:-------------:|:----:|:----:|:------:|:------:|
> |   N:M (A100)  |   2  |   0  |    0   |    0   |
> | **CS (V100)** | 1.39 | 1.86 |  2.48  |  3.07  |
>
> Table 8. ResNet50 speedups comparison at different sparsities on V100 and A100
>
> Q5: Please address the concern in weakness: It seems that the propose CS requires specific optimization of the GPU kernel. It is thus unclear that when other sparsity patterns are optimized on cuda kernel level on a common GPU, whether the proposed CS sparsity can have better speed.
>
> A5: Thanks. For unstructured sparsity, it is impossible to be accelerated on GPUs even with CUDA kernels due to the severe memory conflict. For fine-grained structured sparsity such as OVW, and Shfl\_BW, these works do utilize CUDA kernels for acceleration. Among these sparse patterns, our pattern's advantage is that CS strikes a better tradeoff between CNN accuracy and speedups. That is, under the same speedup through weight sparsity, our CS entails minimal accuracy drops as shown in Figure 6.
>
> Q6: fixed the overlapped "Sparsity" with column line in table 3
>
> A6: Thanks, we have cancelled the overlap in the newly uploaded version.
>
> Q7: table 3, use the same network for all sparsities, so that the insight about trend of accuracy improvements can be obtained.
>
> A7: Thanks. The figure is posted in Appendices as Figure 7. From the figure, it is observed that the curves of our CS are generally above the curves of other structured patterns and are overlapped with curves of unstructured sparsity. That observation further demonstrates the superiority of our CS in terms of accuracy across all the sparsity ratios.

---

> ### Author Response · Authors · 2023-09-17
> **Reply to your comments-3**
>
> Q8: It is surprising that the CS can outperform unstructured sparsity by such a large amount. More explanations and insights are needed.
>
> A8: Thanks. The phenomenon is caused by the small dataset size.
> 1) Generally, compared to modern DNNs' capacity, CIFAR100 is easy to overfitting. Thus, within a certain sparsity range, weight sparsification can largely regularize the model capacity and lead to higher accuracy over dense counterparts. Compared to unstructured sparsity, fine-grained structured sparsity such as CS and N:M constrains a model more strictly. This constraint generally endows the model better regularization on small datasets like CIFAR100 and at low sparsity like 50\% and 75\%. For example, as shown in Figure 7, at 50\% sparsity for VGG19 on CIFAR100, N:M sparsity leads to 73.00\% accuracy, which is better than 72.82\% of unstructured sparsity. At 75\% sparsity, our CS-based VGG19 reaches 72.51\% accuracy, while unstructured sparsity-based VGG19 is 72.13\%.
> 2) On CIFAR100 under high sparsity such as 93.75\%, the accuracy gap between '**CS-C**' and 'unstructured' was narrow due to the small dataset size. Hence, with our proposed training scheme, '**CS(Ours)**' has the possibility to further become on par with or to even surpass 'unstructured'.
> 3) On ImageNet, our CS and other fine-grained sparsity patterns bring constantly lower accuracy than unstructured sparsity regardless of sparsity ratios and network architectures.
>
> Thank you very much for your valuable comments.

---

### Review · Reviewer_w9YE · 2023-09-04

**Summary Of Contributions:**

The manuscript introduces a novel fine-grained pruning technique termed 'complementary sparsity,' aimed at boosting model accuracy while preserving parallelism for enhanced performance on standard CPUs and GPUs. Through comparisons with existing coarse-grained and fine-grained pruning approaches, the authors demonstrate that their proposed method outperforms others in terms of model accuracy, specifically when tested on CIFAR-100 and ImageNet using VGG19 and ResNet architectures. Additionally, the manuscript includes an exploration of relevant hyperparameters."

**Audience:**

Yes

**Claims And Evidence:**

No

**Requested Changes:**

Please refer to 'weakness' above

**Strengths And Weaknesses:**

[Strengths]

- The manuscript provides a comprehensive set of metrics, including sparsity levels, model accuracy, computational speed-up, and FLOPs, to evaluate the proposed pruning method.
- The inclusion of the well-established N:M sparsity scheme in the experiments lends credibility to the study.
- The proposed 'complementary sparsity' technique demonstrates competitive performance across a range of CNN architectures.

[Weaknesses]

- The manuscript lacks an in-depth analysis to explain why the proposed pruning method yields superior model accuracy. Insights into the relationship between the distribution of significant weights and pruning outcomes are notably missing.
- The choice of models for experimentation appears dated. Given that even contemporary CPUs and GPUs—including ARM chips—can efficiently handle models like ResNets, the study should extend its scope to include Transformer-based models or provide theoretical backing to explain its applicability to state-of-the-art architectures.
- The claim that the proposed method surpasses unstructured pruning methods in terms of accuracy warrants further justification. Specific details addressing this point are needed to make such a claim credible.
- Too many typos.

[Overall Remarks]

While the proposed method shows promise in maintaining high levels of parallelism, the empirical results are less convincing due to the selection of older models. For the manuscript to make a more compelling case, it should offer detailed analyses that demonstrate the method's viability for cutting-edge architectures. Additionally, the authors should elaborate on why their approach is particularly well-suited for contemporary parallel computing architectures.

---

> ### Author Response · Authors · 2023-09-18
> **Reply to your comments**
>
> Q1:The manuscript lacks an in-depth analysis to explain why the proposed pruning method yields superior model accuracy. Insights into the relationship between the distribution of significant weights and pruning outcomes are notably missing.
>
> A1: Thanks. We have clarified in main texts and appendices that our CS mainly achieves a comparable accuracy as N:M sparsity. The role of our proposed training scheme is only to improve CS-based CNNs' accuracy under high sparsity like 93.75\%. Instead, a better acceleration affinity on CPUs and common GPUs under comparable accuracy, is truly our CS's advantage over N:M sparsity. Additionally, the phenomenon that the proposed method surpasses unstructured pruning in accuracy on CIFAR100 is caused by the small dataset size, which is detailed in our answer below.
>
> As for CS's higher model accuracy over channel pruning and vector-wise sparse patterns like OVW, we found a metric called mask diversity that may provide some insights. The metric is defined as the number of possible masks for a sparse pattern given a sparsity ratio. For example, for a $8\times 8$ weight tensor and a 50\% sparsity, our CS has $2^{32} = 4,294,967,296$ possible masks, while other vector-wise sparsity such as OVW can only have $ \binom{16}{8} = 12870 $ masks. In this case, not to mention that channel pruning can only provide $ \binom{8}{4} = 70 $ masks. Overall, CS's high mask flexibility probably promotes the high accuracy of CS-based CNNs. For more information, please see reference in our Appendices.
>
> Q2: The choice of models for experimentation appears dated. Given that even contemporary CPUs and GPUs—including ARM chips—can efficiently handle models like ResNets, the study should extend its scope to include Transformer-based models or provide theoretical backing to explain its applicability to state-of-the-art architectures.
>
> A2: Thanks. 1) We validate our CS on DeiT-small as shown in Table 10.
> | Sparsity(%) |   50  |   75  |  87.5 | 93.75 |
> |-------------|:-----:|:-----:|:-----:|:-----:|
> |     N:M     | 77.61 | 73.69 | 67.56 | 60.08 |
> |   CS(Ours)  | 77.4 | 73.66 | 67.63 | 60.01 |
>
> Table 10. Comparison between CS and N:M sparsity
> using Top1 accuracy of DeiT-small on ImageNet. The dense DeiT-small is 75.56\%.
>
> Note to ensure an absolutely fair comparison, CS and N:M -based DeiT-small networks employ identical training settings. Experimental results show once more that our CS incurs a comparable accuracy as N:M sparsity. Note that at 50\% sparsity, both N:M and CS -based DeiT-small surpass the dense one, so the difference between the two sparse patterns at 50\% sparsity is trivial.
> 2) Although some contemporary hardware on cloud sides have held the dense CNNs, CNN sparsification is still meaningful since it can further improve the inference speed and energy-efficiency of modern CNNs on these hardware. Moreover, on some edge sides such as sensors, the existing CNN model compression technologies like sparsification are still not adequate for strict power limit in sensor scenarios. Thus, there is no limit for seeking sparse networks with higher accuracy and less computations. In this work, we try exactly to push the envolope further with our CS pattern.
>
> Q3:The claim that the proposed method surpasses unstructured pruning methods in terms of accuracy warrants further justification.
>
> A3: Thanks. The phenomenon that the proposed method surpasses unstructured pruning methods in terms of accuracy on CIFAR100 is caused by the small dataset size.
> 1) Generally, compared to modern DNNs' capacity, CIFAR100 is easy to overfitting. Thus, within a certain sparsity range, weight sparsification can largely regularize the model capacity and lead to higher accuracy over dense counterparts. The phenomenon was also reported by prior arts, please see Appendices. Moreover, compared to unstructured sparsity, fine-grained structured sparsity such as CS and N:M constrains a model more strictly. This constraint generally endows the model better regularization on small datasets like CIFAR100 and at low sparsity like 50\% and 75\%. For example, as shown in Figure 7 in Appendices, at 50\% sparsity for VGG19 on CIFAR100, N:M sparsity leads to 73.00\% accuracy, which is better than 72.82\% of unstructured sparsity. At 75\% sparsity, our CS-based VGG19 reaches 72.51\% accuracy, while unstructured sparsity-based VGG19 is 72.13\%.
> 2) On CIFAR100 under high sparsity such as 93.75\%, the accuracy gap between '**CS-C**' and 'unstructured' was narrow due to the small dataset size. Hence, with our proposed training scheme, '**CS(Ours)**' has the possibility to further become on par with or to even surpass 'unstructured'.
> 3) On ImageNet, our CS and other fine-grained sparsity patterns bring constantly lower accuracy than unstructured sparsity regardless of sparsity ratios and network architectures.
>
> Q4:Too many typos.
>
> A4: Thanks. We have thoroughly inspected the manuscript and corrected all the typos.

---

### Decision · Action_Editors · 2023-10-12

**Recommendation:** Accept with minor revision

**Comment:**

This paper explores fine-grained structural pruning of deep learning models. The technique, referred to as “complementary sparsity”, build upon the N:M sparsity, but with constrained mask patterns, which is shown to have comparable accuracy performance as N:M sparsity, also enables a parallel accelerating algorithm that can be adapted to common GPUS without sparse tensor cores. The proposed methods are evaluated on a few popular CNN models and CIFAR/ImageNet datasets, demonstrating competitive accuracy and speedup.

Overall, the paper is reasonably well-written. The concept of complementary sparsity is quite straightforward and offers clear advantage on terms of acceleration, without requiring special hardware, enabling its implementation on standard GPUs. The custom GPU kernel implementation and the results demonstrate the hardware speedup. The training algorithm for pruning may not be novel but demonstrates accuracy level comparable to N:M sparsity method.

The reviewers have raised two primary concerns:

1). The evaluation is limited to a small number of relatively older CNN models. While the accuracy performance is evaluated on both CIFAR and ImageNet dataset, the speedup evaluation is primarily on CIFAR100 dataset.

2). Lack of speedup comparison with N:M sparsity.

During rebuttal, the authors presented additional data and clarifications, to some extend addressing these concerns. In particular, the authors provide accuracy performance on small transformer models and speedup comparison with block sparsity and N:M sparsity.

To summarize, this paper is reasonably solid albeit with some reservations. Two reviewers recommend accept, while one reviewer leaning rejection. Considering the proposed sparsity patten is interesting and potential benefits offered by customer kernels for users lacking access to specialized hardware, I am inclined towards acceptance. However, certain modifications are warranted. In particular, the new data and explanations that presented in the rebuttal should be incorporated in the final version with comprehensive explanation. The authors are expected to fulfill their promise of releasing the code and the kernel implementation.  Additionally, the authors should fix typos and figure label issues thoroughly.

**Audience:**

The idea of complementary sparsity is interesting with regard to hardware acceleration. Customer kernels have potential to providing speedup to users who do not have access to specialized hardware for running deep learning models.

**Claims And Evidence:**

The claims are reasonably supported with clear evidence. There are some confusions pointed out by the reviewers, such as claiming better accuracy performance than unstructured pruning, which were addressed by the authors during rebuttal. While the methods were initially evaluated on a limited set of CNN models, the authors provide additional data on small transformer models during rebuttal.